# Aging, NRF2, and TAU: A Perfect Match for Neurodegeneration?

**DOI:** 10.3390/antiox12081564

**Published:** 2023-08-04

**Authors:** Mirjam Brackhan, Marina Arribas-Blazquez, Isabel Lastres-Becker

**Affiliations:** 1Instituto de Investigación Sanitaria La Paz (IdiPaz), 28029 Madrid, Spain; mirjam.brackhan@idipaz.es; 2Instituto de Investigaciones Biomédicas “Alberto Sols” UAM-CSIC, c/Arturo Duperier 4, 28029 Madrid, Spain; 3Department of Pharmacology and Toxicology, Faculty of Veterinary Medicine, Universidad Complutense de Madrid, Avda. Puerta de Hierro s/n, 28040 Madrid, Spain; marina.arribas@vet.ucm.es; 4Instituto Universitario de Investigación en Neuroquímica, Universidad Complutense de Madrid, 28040 Madrid, Spain; 5Department of Biochemistry, School of Medicine, Universidad Autónoma de Madrid, 28040 Madrid, Spain; 6Institute Teófilo Hernando for Drug Discovery, Universidad Autónoma de Madrid, 28029 Madrid, Spain; 7Centro de Investigación Biomédica en Red de Enfermedades Neurodegenerativas (CIBERNED), Instituto de Salud Carlos III, 28031 Madrid, Spain

**Keywords:** tauopathies, oxidative stress, neuroinflammation, neurodegeneration, PiD, CBD, AD, FTLD, DS

## Abstract

Although the trigger for the neurodegenerative disease process is unknown, the relevance of aging stands out as a major risk for the development of neurodegeneration. In this review, we highlighted the relationship between the different cellular mechanisms that occur as a consequence of aging and transcription factor nuclear factor erythroid-2-related factor 2 (NRF2) and the connection with the TAU protein. We focused on the relevance of NRF2 in the main processes involved in neurodegeneration and associated with aging, such as genomic instability, protein degradation systems (proteasomes/autophagy), cellular senescence, and stem cell exhaustion, as well as inflammation. We also analyzed the effect of aging on TAU protein levels and its aggregation and spread process. Finally, we investigated the interconnection between NRF2 and TAU and the relevance of alterations in the NRF2 signaling pathway in both primary and secondary tauopathies. All these points highlight NRF2 as a possible therapeutic target for tauopathies.

## 1. Introduction

The main risk for the incidence of a variety of pathologies, such as neurodegenerative diseases, is aging. As we age, a multitude of changes occur at the molecular level, such as an increase in oxidative stress and a decrease in the antioxidant capacity of our organism, with a decrease in the levels of the transcription factor NRF2 being one of the main causes of this imbalance. This creates a favorable environment for the development of neurodegenerative diseases, such as tauopathies. In this review, we want to highlight the interconnection between aging, the transcription factor NRF2, and TAU.

## 2. The NRF2 Pathway

NRF2, a basic region-leucine zipper (bZip) transcription factor, was originally described as the master regulator of redox homeostasis [1,2]. The main mechanism regulating its transcriptional activity is the control of protein stabilization by the E3 ligase adapter Kelch-like erythroid-cell-derived protein with cap’n’collar homology (ECH)-associated protein 1 (KEAP1) (Figure 1). KEAP1 is a homodimeric protein that links NRF2 with the E3 ligase complex formed by Cullin 3 and RING-box protein 1 (CUL3/RBX1). Under basal conditions, there are low levels of NRF2 due to the action of KEAP1 that binds to and negatively regulates NRF2 [3]. The N-terminal domain of the KEAP1 homodimer binds to one molecule of NRF2 at two amino acid sequences with low (aspartate, leucine, and glycine; DLG) and high (glutamate, threonine, glycine, and glutamate; ETGE) affinity and hence presents NRF2 to ubiquitination by CUL3/RBX1 [4] and subsequent degradation by the proteasome (Figure 1, constitutive conditions).

Since KEAP1 is a redox and electrophile sensor, the modification of critical cysteines by xenobiotics and oxidative or electrophilic stress provoke a conformational change in the NRF2–KEAP1 complex (Figure 1, NRF2 induction). This leads to the loss of affinity to NRF2, which prevents its degradation and allows the accumulation of newly synthesized NRF2, which can then translocate to the nucleus, form heterodimers with small Maf proteins (sMaf), bind to an enhancer sequence termed “antioxidant response element” (ARE) in the promoter regions of NRF2-dependent genes [5], and recruit transcriptional machinery [1]. An alternative mechanism of regulation of NRF2 stability is the phosphorylation mediated by glycogen synthase kinase 3 (GSK-3). This kinase phosphorylates a domain of NRF2 (aspartate, serine, glycine, isoleucine, serine; DSGIS) and hence creates a recognition motif for the E3 ligase adapter β-transducin repeat-containing E3 ubiquitin protein ligase (β-TrCP) that presents NRF2 to the CUL1/RBX1 complex, leading to an alternative pathway for ubiquitin-dependent proteasome degradation of NRF2 [6,7]. Thus, NRF2 protein levels are tightly regulated by KEAP1 and GSK-3/β-TrCP in the context of redox homeostasis and cell signaling, respectively [8]. In addition to these two main mechanisms of NRF2 regulation, the phosphorylation of NRF2 by other kinases has been described as a critical point in the post-translational regulation of this transcription factor [9,10]. For example, it has been described that protein kinase C (PKC) phosphorylates NRF2 in the Neh2 domain, which releases NRF2 from KEAP1 [11,12]. The Neh1 domain is required for the heterodimerization of NRF2 with sMAF, and AMP-activated kinase (AMPK) has been identified to phosphorylate NRF2 in this domain, enhancing NRF2 activity [13,14]. Additionally, the Neh4 and Neh5 domains of NRF2 synergistically participate in the recruitment of NRF2 coactivators and corepressors, modulating the transactivation of NRF2 target genes, and casein kinase 2 (CK2) phosphorylates NRF2 at multiple residues in these two domains [15,16]. The involvement of other kinases in NRF2 phosphorylation, such as protein kinase RNA-like endoplasmic reticulum kinase (PERK), cyclin-dependent kinase 5 (CDK5), and mitogen-activated protein (MAP) kinase, has also been described [17,18,19], although further studies are required to verify the function of these phosphorylations.

## 3. Aging and Its Connection to NRF2

During aging, there is a decline in cellular functions in our organism. The main characteristics of aging at the molecular, cellular, and systemic levels have recently been updated from 9 to 12. This revision of hallmarks includes genomic instability, telomere attrition, epigenetic alterations, loss of proteostasis, disabled macroautophagy, mitochondrial dysfunction, cellular senescence, stem cell exhaustion, chronic inflammation, dysbiosis, nutrient-sensing dysregulation, and altered intercellular communication (Figure 2) [20]. These hallmarks are interdependent since the alteration of one leads to changes in others as well. Concerning the transcription factor NRF2, aging is the main cause of the decline in its activity, since it has been shown that its capacity to induce the expression of antioxidant and detoxifying enzymes decreases with age [21,22,23]. Given that NRF2 is a pleiotropic transcription factor involved in the modulation of more than 250 genes associated with biotransformation (which is the alteration of a substance within the body) and detoxification [24,25], redox homeostasis [26,27], lipid and carbohydrate metabolism [28,29], inflammation [30,31,32,33], epigenetics [34], autophagy [35,36,37], and proteasomes [32,38,39,40] (Figure 3), which are mechanisms involved in aging, there is clearly an interdependence between aging and NRF2. In this review, we will focus on some of these relevant points, such as genomic instability, loss of proteostasis/autophagy, cellular senescence, stem cell exhaustion, inflammation, and mitochondrial dysfunction. 

### 3.1. Genomic Instability and NRF2

The maintenance of genomic integrity is essential for the homeostasis of all cellular functions, including neurons [41]. Neurons are one of the cell types with the highest energy demand and consume approximately 80% of the brain’s energy [42]. Depending on the brain area and its activity, the energy requirements are different. Therefore, neurons display high oxidative metabolic demands, substantially increasing the risk of oxidative damage. As early as the 1950s, Harman had already put forward the theory that oxidative damage to cells could be associated with aging, enhancing the accumulation of damaged macromolecules by oxidative stress and leading to the progressive deterioration of cells [43]. Thus, in relation to neurons, aging is the main cause associated with the progression of neurodegenerative diseases and is associated with genomic instability [44]. One of the main causes of genomic instability is oxidative stress, inducing changes, for example, in DNA bases, mainly at the guanine level, due to its low oxidation potential, leading to the formation of 8-oxo-7,8-dihydroguanine (8-oxoG) [45]. It has been described that 8-oxoG is important for the maintenance and transfer of genetic information into proper gene expression [41]. On the one hand, low basal levels are important for epigenetic modulation of neurodevelopment and synaptic plasticity. On the other hand, increased levels of 8-oxoG damage the genome, leading to somatic mutations and transcription errors. Thus, modulation of oxidative stress at the level of the transcription factor NRF2 is essential to try to maintain cellular homeostasis at the brain level. But as mentioned before, NRF2 levels also decrease during the aging process [21,22,23], thus enhancing genomic instability. In fact, most studies carried out to determine the involvement of NRF2 in genomic instability are concerning cancer, where it has been described that NRF2 preserves genomic integrity by facilitating ataxia-telangiectasia-mutated and RAD3-related (ATR) activation and G2 cell cycle arrest [46] and plays a role in maintaining the integrity of the genome by modulating genes implicated in DNA repair [47].

### 3.2. NRF2 Is a Regulator of Protein Degradation Systems, Proteasomes, and Autophagy

During aging, there is an accumulation of damaged or misfolded proteins that hinder cellular function and tissue homeostasis, associated with the appearance of protein aggregates in neurodegenerative diseases, such as the accumulation of TAU in tauopathies. Damaged or misfolded proteins can be degraded by either the proteasome, macroautophagy (autophagy), or chaperone-mediated autophagy (CMA), and these processes decline during aging [48,49,50,51]. Although the expression of genes associated with proteasome generation are mainly regulated by NRF1 [52], depending on the cell type and state, NRF2 may also be involved [53,54,55], and the absence of NRF2 potentiates aggregation associated with neurodegeneration [32]. In relation to autophagy, NRF2 regulates the expression of the genes *SQSTM1*/p62, *CALCOCO2*, *ULK1*, *ATG2B*, *ATG4D*, *ATG5*, and *GABARAPL1* involved in this process [36], and the induction of NRF2 activity leads to beneficial effects on degeneration through activation of this process [35,56,57]. In addition, more specifically, it has been shown that NRF2 can modulate CMA through the regulation of LAMP2A [37]. This process plays an essential role in TAU degradation, since it has been shown that TAU contains the pentapeptide motif necessary for degradation by this pathway [58] and that acetylated TAU inhibits CMA and promotes TAU pathology propagation [59]. Subsequently, the interconnection between NRF2, autophagy, and TAU and their involvement in neurodegeneration will be discussed in more detail.

### 3.3. Implication of NRF2 in Cellular Senescence and Stem Cell Exhaustion

Cellular senescence is a ubiquitous process, which involves cell cycle arrest, the release of inflammatory cytokines, the exhibition of morphological alterations, and a senescence-associated secretory phenotype (SASP). Several biomarkers, such as β-galactosidase and the cyclin-dependent kinase inhibitors p16, p21, and p53, have been identified [60]. Senescence can be induced by different types of signals, by different kinds of stress, or in response to both intrinsic and extrinsic stimuli [61]. Senescent cells remain viable, with modifications in metabolic activity, and are generally resistant to apoptosis. Regarding cell cycle arrest and p21 upregulation, it has been described that p21-mediated cell protection requires the transcription factor NRF2, indicating a crosstalk between both signaling pathways. Furthermore, there is a direct interaction between p21 and NRF2, which competes with KEAP1 binding [62], compromising the degradation of NRF2 and thus regulating oxidative stress and cellular senescence [63]. However, it has been observed that activation of NRF2 in fibroblasts induces cellular senescence hallmarks, such as the deposit of a senescence-promoting matrix, with plasminogen activator inhibitor-1 being a key inducer of the senescence program. This has been shown in constitutively active NRF2 (caNrf2) mutants lacking a KEAP1-binding domain. Interestingly, these fibroblasts did not show cellular hypertrophy and increased intracellular ROS, other characteristic hallmarks of senescence, highlighting the dual role of NRF2 in the cellular senescence process [64].

Cellular senescence has a major impact on neural stem cells (NSCs). These NSCs within the niches can adopt different states from quiescent, activated, and differentiating, although the proportion of each of these states changes dramatically with aging [65,66]. In particular, senescent cells accumulate during aging, impacting stem cell niches, impairing homeostatic functions, and preventing regeneration. In turn, they also have an impact on the microenvironment of these niches, secreting different pro-inflammatory factors, ROS among others, in a way that affects the surrounding cells, and show a SASP [67,68,69]. The involvement of NRF2 in the maintenance of these niches and NSCs highlights the importance of this transcription factor in this whole process because the regulation of the redox balance is an essential determinant in the state and function of NSCs. The fact of NRF2 decline with aging interrelates these three events. Indeed, it has been described that NRF2 acts as a key pluripotency gene in human embryonic stem cells (hESCs) and that its expression is highly enhanced in hESCs and declines upon differentiation [70]. Within the same study, it has been described that NRF2 inhibition impairs both the self-renewal ability of hESCs and the reestablishment of pluripotency during cellular reprogramming. This point is important because in the adult brain, there are two regions of lifelong neurogenesis: the subventricular zone (SVZ) of the striatum and the subgranular zone (SGZ) of the hippocampus. It has been described that reduced expression of NRF2 mediates the decline in NSC function in the SVZ of the striatum during aging [71]. However, it has been reported that NRF2-deficient mice exhibited impaired long-term potentiation at the SGZ level, due to a reduction in NSCs from birth to adulthood. Furthermore, the lack of NRF2 impaired neuronal differentiation in the SGZ, leading to abnormal production of astrocytes, neurons, and oligodendrocytes [72]. The relevance of NRF2 is focused not only on NSCs but also on STEM cells in general, which has been extensively reviewed [73,74,75].

### 3.4. NRF2 as an Inflammation Modulator

In the central nervous system (CNS), apart from neurons, essential functions are performed by astrocytes and microglia implicated in neuroinflammation, which are also influenced by aging processes. Astrocytes, the most abundant glial cells within the CNS implicated in glutamate homeostasis and neurogenesis processes, are the metabolic and energetic support of neurons and are involved in synapse assembly and function, forming part of the tripartite synapse, among many other functions. During aging, these cells undergo changes in gene expression, leading to the appearance of senescent forms of astrocytes [76]. In contrast, microglial cells are the specific macrophages of the CNS that are specialized in phagocytizing apoptotic neurons, eliminating aggregated extracellular proteins, defending against infections, and pruning synapses, among other functions [77]. Aging also affects the function of these cells by producing changes in their metabolism and oxidative stress, enhancing the gene heterogeneity of microglial cells, with differences between brain regions [78]. It has been described that in astrocytes, there is poor mitochondrial respiration but high ROS production, whereas neurons have high mitochondrial respiration and low ROS production [79,80]. Therefore, it could be likely expected that the NRF2 signaling pathway could be predominantly implicated in astrocytes/microglia, and its alteration is associated with changes in the neuroinflammatory process [31,32,35], although there is controversy in the field as to whether NRF2 activation only occurs at the glial cell level or whether neurons may also be involved [81]. Furthermore, NRF2 has been described to be involved in the modulation of neuroinflammatory processes by its crosstalk with the transcription factor nuclear factor of κ-light chain of enhancer-activated B cells (NF-κB), a master modulator of the inflammatory response [82]. The interconnection between both transcription factors is supported by the fact that in the absence of NRF2, there is an exacerbation of the NF-κB-mediated inflammatory process. Furthermore, in the promoter region of *NEF2L2*, there is a κB region, and KEAP1 can interact with and regulate the inhibitor of nuclear factor κB kinase subunit β (IKKβ), modulating the NF-kB pathway [10]. The modulation of NRF2 in response to NF-κB activation can act as a protective mechanism against inflammation through the small GTPase RAC1 [82,83], and the expression of pro-inflammatory genes is affected (Figure 3). All this evidence converges in the fact that during the aging process, neuroinflammation is enhanced by the senescence of astrocytes and microglia, decreased expression of NRF2, and less inhibition of the NF-κB pathway. Another possible mechanism by which NRF2 could be regulating neuroinflammatory processes would be through microRNAs (miRNAs), small non-coding RNA molecules that regulate the activity of approximately half of all protein-coding genes [84]. Recently, it has been described that NRF2 is a modulator of miRNA biogenesis [34]. It has been suggested that several miRNAs directly repress the post-transcriptional expression of NRF2 and thereby negatively regulate the NRF2-dependent cellular cytoprotective response [85]. Thus, the modulation of neuroinflammatory processes by miRNAs and NRF2 is a mechanism yet to be explored, as there are no concrete studies that have described this relationship in relation to tauopathies.

## 4. TAU and Its Modulation during Aging

The microtubule-associated protein TAU encoded from the *MAPT* gene, in the adult human brain, has six different isoforms resulting from alternative splicing of exons 2, 3, and 10 [86,87]. The differences between the isoforms stem from the presence or absence of two inserts of 29 amino acids encoded by exons 2 and 3, referred to as TAU 0N, 1N, or 2N, and the presence or absence of the R2 domain, one of the four partially repeated microtubule-binding domain regions (MTBRs) designated as R1, R2, R3, and R4 (Figure 4) [88,89,90]. In addition, *MAPT* contains a common inversion polymorphism leading to the presence of two different haplotypes referred to as H1 and H2 haplotypes [91]. Interestingly, genome-wide association studies (GWAS) have shown that the *MAPT* H1 haplotype is associated with several tauopathies, as we will describe later [92,93]. The function of the TAU protein is not yet fully understood. It is known that it facilitates the assembly of tubulin into microtubules by stabilizing them [94], although it cannot be ruled out that it may have other functions that are still unidentified [95]. Through posttranslational modifications (PTMs), mainly phosphorylation, the TAU protein can have its functionality altered and become prone to aggregate. It has been described that TAU aggregation accumulated in an age-dependent manner in a *Caenorhabditis elegans* (*C. elegans*) model [96]. In contrast, in human brain samples, it has been described that total soluble TAU levels declined with age [97]. As a consequence of aging, the accumulation of damage generated by oxidative stress, directly or indirectly through mitochondrial alterations, can produce modifications in TAU that facilitate its aggregation [91,94], which has been associated with the development of neurodegenerative diseases, like Alzheimer’s disease (AD). However, TAU can be secreted into the extracellular space, spreading the protein to surrounding cells [98], such as other neurons, astrocytes, and microglia, in a prion-like manner. In particular, an age dependence of TAU protein spread in mouse brains has been observed. Old animals showed more TAU spreading in the hippocampus and adjacent cortical areas and accumulated more misfolded TAU in entorhinal cortex neurons [99]. All this evidence indicates that changes in TAU protein levels, PTMs, and spreading occur during aging.

## 5. Aging, Mitochondrial Dysfunction, and Ferroptosis: Role of NRF2 and TAU

It has been described that gradual mitochondrial dysfunction occurs with aging (Figure 2). It is recognized that mitochondria not only play the role of energy suppliers in the cell but also play an essential role in regulating cellular metabolism and homeostasis due to their key roles in bioenergetics, generation of ROS, anabolism and catabolism, iron–sulfur cluster and heme biosynthesis, calcium and iron homeostasis, apoptosis, and signal transduction [102,103]. Therefore, their dysregulation is strongly related to aging and neurodegenerative processes [104]. In relation to NRF2, it has been described that NRF2 deficiency leads to mitochondrial depolarization, decreased ATP levels, and impaired respiration. Mitochondrial oxidation of long-chain (palmitic) and short-chain (hexanoic) fatty acids is repressed in the absence of NRF2 [105]. In addition, the activity of respiratory complexes is also severely impaired, indicating that NRF2 directly regulates cellular energy metabolism [106]. Furthermore, NRF2 is activated by the disruption of mitochondrial thiol homeostasis but not by enhanced mitochondrial superoxide production, suggesting that alterations in the mitochondrial redox homeostasis can be sensed differentially within the cell [107]. All these data point to an essential role of NRF2 in mitochondrial function and therefore its implication in the neurodegenerative process [108,109]. In relation to tauopathies, or more precisely to the TAU protein, it has been described that abnormal TAU impairs mitochondrial function, although the exact mechanisms are still unknown [110]. One possible mechanism is related to the fact that TAU is involved in cargo transport along the axon, including the mitochondria. Alterations in the TAU protein could lead to a decrease in mitochondrial transport to the synapse, finally resulting in synaptic degeneration and neuronal death [111]. In addition, a reduction in the number of mitochondria in axons has been found with TAU^P301L^ in the rTg4510 transgenic mouse model. Perinuclear clustering of mitochondria, elongation of the mitochondrial network, and disruption of mitochondrial permeability transition pores were also observed in relation to TAU alterations [112,113,114]. Also, more recently, it has been demonstrated that TAU is localized within mitochondrial sub-compartments and impacts mitochondrial physiology [115]. All this evidence reveals the role of both NRF2 and TAU in the mechanisms associated with mitochondrial function.

One common feature of neurodegenerative diseases is mitochondrial dysfunction and overproduction of ROSresulting in oxidative stress [116], and lipid peroxidation is one of the markers of oxidative stress. The products of lipid peroxidation as well as RAS-selective lethal small molecules erastin and (1S,3R)-RSL3 (RSL3) have been shown to be the trigger for ferroptosis. Erastin blocks the uptake of cystine through system xc− and depletes cellular glutathione (GSH) [117,118]. Deficiency in the well-defined NRF2 target genes solute carrier family 7 member 11 (SLC7A11; a subunit of the cystine/glutamate antiporter xCT) and the catalytic and modifier subunits of glutamate cysteine ligase (GCLC/GCLM), both of which control the level of GSH (a cofactor for GPX4), can lead to the initiation of ferroptosis, or enhanced susceptibility of cells to pro-ferroptotic agents [119,120]. That is why it has been suggested that NRF2 has anti-ferroptotic activity. Related to TAU, there are not many studies that have analyzed the implication of ferroptosis in the alterations associated with this protein. *In vitro* studies have shown that erastin-induced ferroptosis can promote TAU hyperphosphorylation and aggregation in mouse neuroblastoma cells (N2a cells) and that ferrostatin-1, a ferroptosis inhibitor, can alleviate TAU aggregation effectively [121]. Furthermore, it has been described that astrocytes are the major cell type accumulating iron in the early affected regions of progressive supranuclear palsy (PSP), highly associated with cellular TAU pathology [122]. Moreover, in PSP, unlike in Alzheimer’s disease (AD), lipid peroxidation is selectively associated with neurofibrillary tangle formation, suggesting a mechanism that may contribute to hamper TAU degradation, leading to its aggregation in the PSP-specific abnormal filaments [123]. Regarding familial AD presenilin mutations, loss of the presenilins dramatically sensitizes multiple cell types to ferroptosis, without TAU implication [124]. Although there is much evidence of the involvement of the ferroptosis process in AD [125,126,127], the relationship between amyloid precursor protein (APP), TAU, and ferroptosis has not yet been described. It has been found that APP co-localizes with the divalent channel ferroportin 1 (FPN1) and that TAU transports APP to the cell membrane and stabilizes the FPN1–APP complex. How this mechanism is deregulated in AD and the involvement of the TAU protein remain to be discovered.

## 6. NRF2 and TAU Interconnection

In recent years, the increased discovery of genes regulated by NRF2 implies that this transcription factor is involved in a multitude of physiological processes, and the fact that NRF2 and TAU are involved in common events makes a relationship between the two proteins feasible. The regulation of TAU levels by NRF2 has been analyzed at the gene and protein levels by two different studies. A study performed by integrating genome-wide maps of NRF2/sMAF analyzed the occupancy level of disease susceptibility loci and found that one of the SNPs (rs242561) was located within the regulatory region of the *MAPT* gene [101]. This SNP, with a significant allele-specific binding, has been predicted to affect NRF2/sMAF binding and transactivation, and the T allele has been foreseen to confer higher affinity for ARE than the C allele (Figure 4). This study hypothesizes that this T allele H2 haplotype, which has been observed previously [128,129,130,131], comprises one of the TAU isoforms containing exon 3. The inclusion of exon 3 in the TAU protein has been described to be protective as it decreases its tendency to aggregate. Thus, individuals carrying rs242561 with the T allele in the *MAPT* gene should benefit from increased resistance to TAU aggregation, especially under conditions of oxidative stress, where activation of the NRF2 pathway also occurs [101]. However, it has been described that activation of the NRF2 pathway can reduce the levels of phosphorylated TAU, mainly by modulating one of the autophagy receptors, NDP52 [132]. This fact is further enhanced by the finding that NRF2 is able to regulate genes involved in the process of autophagy, as described before [36]. Induction of the expression of proteins involved in autophagy through NRF2 activation facilitates the clearance of phosphorylated TAU and, thus, reduces its aggregation. Furthermore, it has been observed that optineurin (OPTN) selectively targets mainly soluble TAU expressed under physiological conditions, while SQSTM1 predominantly degrades insoluble but not soluble mutant TAU [133]. Interestingly, the expression of NRF2-dependent autophagy-dependent adaptor proteins clears TAU in an age-dependent manner, confirming once again the association between aging, NRF2, and TAU [134]. To date, no studies have focused on determining the exact mechanism by which NRF2 and TAU are interconnected, beyond those we have described here. A possible mechanism by which TAU overexpression induces NRF2 activation will be addressed in the frontotemporal dementia (“Relevance of NRF2 in FTLD”) section. However, as it is the TAU protein with the P301L mutation, we do not know whether this effect is due to the P301L mutation or the overexpression of the TAU protein itself, which would be interesting to investigate. Another determining question is to identify whether this interaction between TAU and NRF2 occurs in the same cell type or in different cell types as a consequence of paracrine or juxtacrine signaling mechanisms. There is increasing evidence that NRF2 activation in neurodegenerative processes occurs at the glial cell level [81], but it remains to be determined in which cell type the interconnection between TAU and NRF2 occurs in each specific tauopathy. As will be explained in the section on each tauopathy, the TAU protein can aggregate in different cell types, such as neurons and astrocytes, which could include diverse mechanisms of action between TAU and NRF2. In addition, it would also be interesting to find out how the possible interconnection between TAU and NRF2 could control cellular metabolism in the aging brain.

## 7. Types of Tauopathies and Their Main Molecular Characteristics

In general terms, tauopathies describe a wide range of phenotypically diverse neurodegenerative diseases whose common feature is the appearance of hyperphosphorylated TAU aggregates [135]. These variations in TAU lesions are associated with morphological diversity, isoform content, cell types involved, and clinical manifestations. The term “tauopathy” was first described by Spillantini in 1997 [136], and since then, these diseases have been classified into primary, secondary, and geographically isolated tauopathies (Figure 5). The term “primary tauopathy” refers to disorders in which intracellular inclusions of the TAU protein are the predominant feature [137]. Secondary tauopathies are diseases that are also pathologically characterized by intracellular TAU aggregates but are driven by a force other than TAU pathology, as is the case for AD [138] and Down syndrome (DS) [139]. In contrast, the etiopathogenesis of geographically isolated tauopathies is still unknown, although environmental impact seems to be highly relevant [140]. Another classification system for tauopathies is based on the TAU isoforms constituting intracellular inclusions [141]. In the adult brain, the 3R and 4R isoforms of TAU are expressed in a 1:1 ratio, whereas in many tauopathies, there is an imbalance with an increase in either the 3R forms, as in Pick’s disease, or 4R, which occurs in most tauopathies (Figure 5).

To date, more than 60 mutations have been identified in the *MAPT* gene (for review [142]). Many of these mutations, such as G272V, L284L, P301L, P301S, V337M, K369I, G389R, R406W, and N410H, reduce the affinity of TAU to microtubules and reduce its ability to stimulate microtubule assembly, while some of them can alter the function of TAU with other proteins or facilitate its phosphorylation, which hinders axonal transport. These mutations are associated to a greater or a lesser degree with the development of diverse tauopathies, although other genes may also be involved. In general, most of the cases of tauopathies are sporadic, and approx. 31% have been reported as a familiar form (with variations within the different tauopathies) [143]. Moreover, most cases presumably have a multifactorial etiology. Within this multifactorial etiology, TAU-associated degeneration appears to play a relevant role in initiating inflammation-related pathology [144,145], mitochondrial dysfunction [146,147,148], and oxidative stress [149,150,151]. Altogether, these processes significantly contribute to the disruption of the redox balance, increasing the formation of ROS and/or promoting dysregulation of the endogenous antioxidant systems involving NRF2. Therefore, we will now analyze the involvement of the NRF2 transcription factor in the onset or progression of tauopathies.

## 8. Primary Tauopathies and Their Connection to NRF2

We will provide an overview of the current evidence supporting the dysregulation and relevance of the transcription factor NRF2 in the development and progression of primary tauopathies. We will focus on those disorders for which there is scientific evidence, either with human samples or with *in vitro* or *in vivo* models, such as FTLD, PiD, (PSP), and corticobasal degeneration (CBD).

### 8.1. Relevance of NRF2 in FTLD

Frontotemporal dementia (FTD) or frontotemporal lobar degeneration (FTLD) covers a whole spectrum of neurodegenerative disorders that principally affect the frontal and temporal lobes of the brain, followed by hippocampal atrophy [152]. There are three clinical variants, including behavioral-variant frontotemporal dementia (bvFTD), non-fluent-variant primary progressive aphasia (nfvPPA), and semantic-variant primary progressive aphasia (svPPA). Approximately half of the FTD spectrum cases are caused by TAU pathology (FTLD-TAU) [142], with a 3R, 3R + 4R, and 4R underlying pathology. Most of the FTLD-TAU cases are sporadic, although rare mutations in *MAPT* (P301L or P301S) cause autosomal dominant FTLD [153,154,155,156,157]. Furthermore, frontotemporal dementia with parkinsonism linked to chromosome 17 (FTDP-17) is also considered a familial form of FTLD-TAU.

Cellular and mouse models of FTLD-TAU expressing mutated forms of human TAU have provided relevant information about the implication of NRF2 signaling in the pathology. At the neuronal level, overexpression of TAU^P301L^ in the hippocampal line HT22 does not induce activation of the antioxidant activity pathway, analyzed with heme oxygenase-1 (HO-1) levels, although it does produce a decrease in p62 levels [31]. Concerning these results, it has been described that in primary cultures of neurons from MAPT^P301S^ transgenic mice, there is a decrease in NRF2 levels and its translocation to the nucleus is inhibited [158]. This study demonstrated that MAPT^P301S^ inhibits NRF2 signaling by acetylating KEAP1 and inhibiting its degradation and decreases the levels of synaptic proteins post-synaptic density protein 93 (PSD93), PSD95, and SYN1, as well as the mRNA levels of their encoding genes *DLG2*, *DLG4*, and *SYN1* [158]. Since *DLG2*, *DLG4*, and *SYN1* were found to contain AREs for NRF2 binding, the authors concluded that MAPT^P301S^ might drive synaptic toxicity via NRF2 inhibition. *In vivo*, they showed that overexpressing NRF2 ameliorates MAPT^P301S^-induced memory loss due to the regulation of these synaptic proteins [158]. In another study, overexpression of TAU^P301L^ in a mouse model based on intrahippocampal injection of an adeno-associated viral vector showed aggravated astrogliosis, microgliosis, and inflammation in the hippocampus of NRF2-deficient mice [31]. In this model, TAU^P301L^ overexpression induced activation of the NRF2 pathway mainly at the level of astrocytes and microglia. Treatment with NRF2 inducers, such as dimethyl fumarate (DMF) or sulforaphane (SFN), was shown to have anti-inflammatory and neuroprotective effects in this model [159,160]. In agreement with these results, in the MAPT^P301S^ tauopathy mouse model, exacerbation of age-dependent reactive changes in astrocytes was described, where NRF2 targets were part of the core signature of astrocytic genes regulated by TAU [161]. Furthermore, the authors showed that astrocyte-specific NRF2 expression induces a reactive phenotype, which recapitulates elements of this signature, reduces phospho-TAU accumulation, and rescues neurodegeneration and cognitive decline. Concerning microgliosis, it has been described that TAU^P301L^ overexpression induces the neuronal release of the chemokine fractalkine (CX3CL1) that modulates NRF2 microglial responses *in vitro* and *in vivo*. NRF2- and CX3CR1-knockout mice did not express HO-1 in microglia and exhibited increased microgliosis and astrogliosis in response to neuronal TAU^P301L^ expression, demonstrating a crucial role of the fractalkine/NRF2/HO-1 pathway in attenuating the pro-inflammatory phenotype in these FTLD-TAU models [31,159,162]. Additionally, another study found that NRF2 deficiency accelerates the development of hind-limb paralysis and leads to mild memory deficits in MAPT^P301S^ mice. However, NRF2 loss did not alter markers of senescence in the brain or increase brain levels of TAU or phosphorylated TAU in this model [163].

All these data suggest that in relation to FTLD-TAU models, NRF2 predominantly plays a role related to the processes of astrocytosis and microgliosis and that NRF2 activation is an attempt by glial cells to control the inflammatory process and thus the progression of neurodegeneration.

### 8.2. Involvement of NRF2 in Pick’s Disease

Pick’s disease (PiD) is a type of bvFTD and can only be diagnosed postmortem, so it is a strict neuropathology term. It is characterized by neuronal loss in the frontal and temporal lobes and aggregation of 3R TAU in pathognomonic inclusions, termed “Pick’s bodies.” These are mainly located in pyramidal CA1 neurons and granular neurons in the dentate gyrus of the hippocampus and in layer II of the frontal and temporal cortex [142,164]. PiD usually presents sporadically. However, various mutations have been identified in exons 9 to 12 of the *MAPT* gene, more specifically K257T, G272V, ΔK280, S320F, K369I, and, most recently, Q336H/R [165,166], in patients with familial PiD. PiD is the only known primary 3R tauopathy, and due to its rarity, it remains significantly understudied. As mentioned before, at the neuroanatomical level, the hallmark of PiD is the presence of swollen neurons termed “Pick cells” and intracellular TAU aggregates known as “Pick bodies”; however, little is known about the exact composition of these bodies.

There are not many studies that have analyzed the involvement of oxidative stress, neuroinflammation, or directly NRF2 concerning this pathology. In samples of patients with PiD, it has been observed that both Pick cells and more precisely Pick bodies are positive for HO-1, one of the main antioxidant enzymes regulated by NRF2. These results suggest, on the one hand, that oxidative stress must indeed be occurring, and, on the other hand, that this enzyme seems to be sequestered in Pick bodies [167]. Subsequently, a more direct study on NRF2 levels and localization has also been performed in samples from patients with Pick’s disease. It was observed that there is a significant decrease in NRF2 protein levels in the frontal cortex, while in the occipital cortex, they remain unchanged. Using immunohistochemical techniques, it was observed that NRF2 is present in the nucleus in both neurons and glial cells still present in areas of the cortex in the early stages of the disease. These data suggest that although there is a significant global decrease in NRF2 in the frontal cortex, the remaining cells continue to activate NRF2 in response to oxidative stress [168]. These data suggest that NRF2 pathway activators, such as DMF or SFN, could be a therapeutic strategy for these patients.

### 8.3. Role of NRF2 in PSP and CBD

PSP and CBD belong to a group of neurodegenerative diseases called parkinsonian syndromes, which also include Parkinson’s disease (PD), dementia with Lewy bodies (DLB), and multiple system atrophy (MSA). Growing evidence suggests that parkinsonian syndromes may have common genetic bases, and to date, two essential factors have been identified, the TAU protein and α-synuclein. Both PSP and CBD are clearly associated with alterations in the TAU protein, identified as 4R tauopathies [169]. PSP and CBD are rare neurodegenerative disorders (prevalence of 3.1–6.5/100.000 and 6/100.000, respectively), share clinical features, and are characterized by short average survival. There is a notable overlap in their clinical presentations, pathological features, biochemistry, and genetic risk factors, and a definitive diagnosis can only be confirmed postmortem [170,171]. From the pathological point of view, both PSP and CBD have neuronal and glial lesions that are composed primarily of hyperphosphorylated TAU, although the distribution patterns differ [172]. Although the majority of PSP and CBD cases are sporadic, certain *MAPT* mutations can result in clinical phenotypes and pathological features that are indistinguishable from PSP or CBD and may be considered the monogenetic causes of PSP and CBD [173]. PSP pathology is associated with some rare *MAPT* mutations, mainly in exon 10 (S285S, DN296, N297K, G303V, S303S, S305S) but also in intron 10 (IVS10 + 16) and exon 1 (R5L) or even in exon 12 (V363A) or exon 13 (R406W). CBD pathology is associated with *MAPT* mutations in exon 10 (N296N, P301S, S305S), intron 10 (IVS10 + 16), exon 11 (K317M), and exon 13 (R406 W, N410H) [142,174].

The relationship between PSP, CBD, and NRF2 has been identified in a study combining genome-wide maps of NRF2/sMAF occupancy with disease-risk SNPs identified in GWAS [101]. As explained in detail before, this study by Wang et al. (2016) suggested the regulation of the *MAPT* gene by NRF2/sMAF (T allele rs242561) in the H2 haplotype, resulting in the production of a TAU protein that resists aggregation. This protective allele showed complete linkage disequilibrium, i.e., a non-random association with the protective G allele of rs8070723 that confers lower risk for PSP and CBD in Europeans, suggesting the potential use of NRF2 pathway activators in disease prevention [101]. This relationship was also observed in analysis of postmortem brain tissue from patients with PSP displaying increased levels of NRF2 [31,175] and antioxidant enzymes, such as HO1, SOD1, SOD2, and GPx [31,176]. These findings suggest that activation of NRF2 signaling is presumably indicative of an attempt to increase cell survival. In the same line of evidence, in relation to CBD, it has been observed that there is an increase in HO-1 levels, suggesting that oxidative stress must indeed be occurring [167].

Taken together, these findings point to activation of NRF2 in PSP and CBD in an attempt to reduce the damage associated with TAU pathology. However, levels of NRF2 activation might be insufficient to control both pathologies, and thus, PSP and CBD patients might benefit from a more pronounced NRF2 response induced by activators of the NRF2 pathway.

## 9. Secondary Tauopathies and Their Link to NRF2

Next, we are going to focus our attention on the connection between NRF2 and diseases where TAU alterations occur as a consequence of other factors, such as AD, DS, and chronic traumatic encephalopathy (CTE).

### 9.1. Implication of NRF2 in AD

Alzheimer’s disease (AD) is the leading cause of dementia worldwide, responsible for 60–80% of cases, and is a progressive disease that begins with mild cognitive impairment that progresses to loss of memory and thinking due to hippocampal and cortical atrophy [177,178,179,180,181]. AD is a multifactorial, complex disease characterized neuroanatomically by the presence of plaques of extracellular amyloid-β (Aβ) and neurofibrillary tangles of hyperphosphorylated TAU protein [182]. However, the exact mechanisms responsible for AD pathogenesis remain elusive. Several lines of evidence suggest that the accumulation of Aβ due to an imbalance between Aβ production and Aβ clearance may be key in initiating the pathological process, as proposed by the amyloid cascade hypothesis. However, aggregated, hyperphosphorylated TAU has been proposed to be a principal driver of neurodegeneration because the rate of TAU accumulation predicts the onset of cognitive impairment [183,184]. Interestingly, the *MAPT* H1 haplotype, which has been linked with several primary tauopathies, as described before, has also been identified as a risk factor for AD [93].

In relation to the transcription factor NRF2, its involvement in AD has been studied at different levels. A case–control study analyzed single-nucleotide polymorphisms and haplotypes of *NFE2L2* for associations with disease risk and AD onset in blood and brain tissue. While *NFE2L2* was not associated with disease risk, common variants of the *NFE2L2* gene may affect disease progression, potentially altering clinically recognized disease onset [185]. Moreover, in blood samples from AD patients at low Braak stages, a transcriptomics study showed that NRF2 is significantly disturbed in these patients [186]. In another study performed on the cerebrospinal fluid (CSF) from patients with mild cognitive impairment and AD, the presence of both polymorphic alleles for *NFE2L2* and *KEAP1* and pro-inflammatory markers was analyzed. Significant associations with cognitive test scores were observed for *KEAP1* rs1048290 and rs9676881, as well as *NFE2L2* rs35652124. In the AD group, KEAP1 remained associated with cognitive scores, indicating that polymorphisms in antioxidative genes might be associated with CSF biomarkers and cognitive test scores [187]. At the mRNA level, in a meta-analysis with seven microarray datasets of AD studies to explore functional enrichment analysis, it was found that NRF2 is upregulated, although its target genes were all downregulated due to overexpressed small MAFs, which acted as transcriptional repressors [188]. Similar results were found in another study using human frontal and temporal lobe samples from AD patients, where increased *NEF2L2* mRNA levels were reported [189].

At the protein level or localization, several studies have addressed whether NRF2 is altered in AD patients’ brain tissue. Assessment of NRF2 in hippocampal CA1 neurons using immunostaining revealed that NRF2 is predominantly localized in the cytoplasm and nearly absent in the nucleus in AD patients. NRF2 did not co-localize with neurofibrillary tangles or plaques in AD patients’ hippocampus. Furthermore, immunoblotting analysis demonstrated reduced NRF2 protein levels in the nuclear fraction of AD patients’ midfrontal cortex [190], indicating that NRF2-mediated transcription is not induced in neurons in AD. In another study, NRF2 levels were analyzed in the total lysate of low-Braak-stage AD patients’ hippocampus. Although in this case, both NRF2 and HO-1 levels were found to be significantly increased in these patients, subsequent more detailed studies at the cellular level suggested that NRF2 is induced at glial cells (astrocytes and microglia) [31]. Further studies are required to determine in which cell type NRF2 is expressed in AD, since to date, the cell-type-specific requirement of NRF2 expression in Alzheimer’s disease is unknown [81]. However, it is possible to speculate that the increase in the NRF2 signaling pathway predominantly takes place in astrocytes/microglia, as is the case in other pathologies, such as Parkinson’s disease [32,81].

To further determine the involvement of NRF2 in the onset and development of AD pathology, several studies have been performed in transgenic mice that combine amyloidopathy and tauopathy. First, it was revealed that the absence of NRF2 in the mouse brain replicates transcriptomic changes found in the human elderly and in AD patients’ brain [191]. Furthermore, the same study showed that in a mouse model that combined amyloidopathy and tauopathy (APP^V717I^/TAU^P301L^), NRF2 deficiency led to increased markers of oxidative stress and neuroinflammation as well as higher levels of insoluble phosphorylated-TAU compared to the corresponding mice that did express NRF2. In contrast, studies performed on another transgenic model of AD carrying a homozygous *PSEN1^M146V^* knock-in mutation and homozygous mutant transgenes for Swedish *APP^KM670/671NL^* and *MAPT^P301L^* (3xTg-AD mice) have not observed differences in the levels of *Nfe2l2* or its signaling pathway [189]. Finally, studies on the possible brain-protective effects of NRF2 inducers on AD pathology in these transgenic models have also been performed. Long-term treatment with DMF reduced neuroinflammation and improved cognitive as well as motor disability in the APP^V717I^/TAU^P301L^ mouse model [192]. Beneficial effects have also been observed for the treatment with SFN, which ameliorated cognitive deficits in 3xTg-AD mice [193]. These observations suggest that pharmacological modulation of NRF2 could be a beneficial therapeutic strategy in the fight against neurodegeneration in AD. Although NRF2 activators are currently being explored as potential therapeutic agents for a variety of diseases [194,195], to date, only one clinical trial of the NRF2 activator sulforaphane has been registered for the treatment of patients with prodromal-to-mild AD (NCT04213391).

### 9.2. Role of NRF2 in DS

Down syndrome (DS) is the most common genetic cause of intellectual disability, affecting approximately 6 million people worldwide, and is associated with accelerated aging and cognitive decline [196,197,198]. Individuals with DS display the two major neuropathological characteristics of AD, i.e., Aβ plaques and TAU neurofibrillary tangles, by the age of 40 years and have an 80% risk of developing dementia at the age of 65 years [139,199]. Early onset of AD pathology in DS is in part caused by the overproduction of Aβ due to the presence of three copies of the gene encoding APP [139,200]. While Aβ pathology might play an important role in driving TAU accumulation in DS, additional triplication of the *DYRK1A* gene encoding dual-specificity tyrosine phosphorylation-regulated kinase 1A might also favor TAU aggregation, due to its implication in TAU phosphorylation [151]. Therefore, as in the case of AD, the involvement of NRF2 may be critical in the development and progression of neurodegeneration in DS.

To date, few studies have addressed the implication of NRF2 in DS. In contrast to what is observed in the brains of AD patients, a significant decrease in NRF2 total protein levels has been shown in the brain from both DS and DS-AD individuals compared with age-matched controls [201]. Peripheral blood mononuclear cells from DS individuals exhibited no changes in total NRF2 protein levels, while lymphoblastoid cell lines from DS patients displayed reduced protein levels of total and nuclear NRF2, as well as its downstream target HO-1 [201,202]. These data seem to point to a decrease in the NRF2 pathway in individuals with DS. One of the genes affected by gene triplication in DS is basic leucine zipper transcription factor 1 (BACH1), a bZip protein belonging to the cap’n’collar family, which is located at Chr21q21.3. Similar to NRF2, BACH1 binds to AREs in antioxidant genes in the nucleus and functions as a transcriptional repressor negatively regulating the expression of *HMOX1* and *NQO1* genes [203,204,205,206]. It has been suggested that the balance between NRF2 and BACH1 in the nucleus determines up- or downregulation of ARE-mediated gene expression [206]. Triplication of the *BACH1* gene on chromosome 21 in DS could shift this balance toward increased BACH1 expression, leading to the displacement of NRF2 and repressed induction of antioxidant response genes [207]. In fact, elevated BACH1 mRNA and protein levels were detected in the frontal cortex of DS subjects both before and after the development of AD neuropathology compared to age-matched controls [201,208]. Correspondingly, lymphoblastoid cell lines from DS individuals exhibited augmented nuclear BACH1 protein levels [201,202]. This could be the main reason for the differences observed between AD patients and individuals with DS, and therefore, the induction of the NRF2 pathway may not be equally effective in both pathologies.

### 9.3. Role of NRF2 in CTE

Chronic traumatic encephalopathy (CTE) is a neurodegenerative disease exhibiting a distinct pattern of neuropathological changes associated with the cumulative impact of repetitive mild traumatic brain injury, leading to increased risk of long-term memory and cognition issues [209,210]. Repetitive head trauma injuries lead to neuronal damage, microhemorrhages, and loss of blood–brain barrier integrity, which triggers TAU protein phosphorylation, among a variety of molecular events [211]. No analyses of NRF2 levels or its signaling pathways have been performed in CTE, but there is speculation that the NRF2 inducer DMF could be beneficial for these patients via its dual action, through modulation of GSK-3 activity and thus TAU phosphorylation, and by promoting the antioxidant activity of the NRF2 pathway [209,210]. However, this will require studies to support this hypothesis.

## 10. Conclusions

In this review, we addressed three events that converge in neurodegenerative processes—aging, the transcription factor NRF2, and the TAU protein—and how they are interconnected with each other. Although in several pathologies, the underlying molecular events are not yet fully understood, the interconnection between NRF2 and TAU is common in both primary and secondary tauopathies, all of which are associated with aging.

## Figures and Tables

**Figure 1 antioxidants-12-01564-f001:**
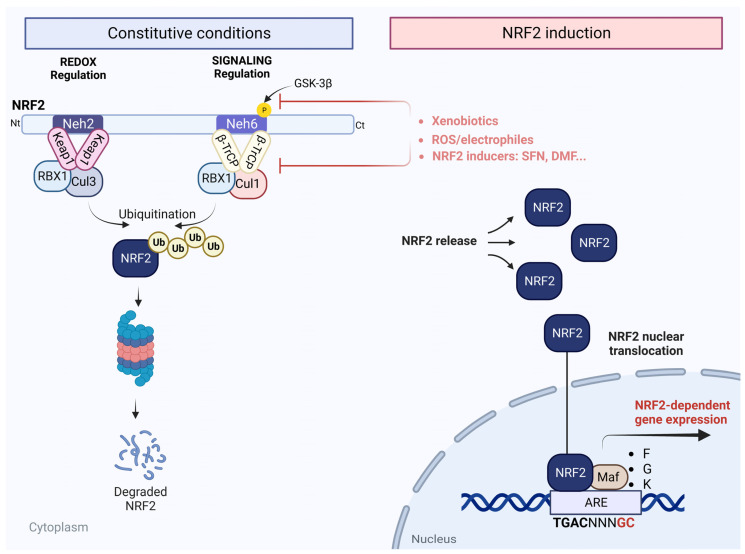
**Scheme of the NRF2 signaling pathway.** Dual-regulation model of NRF2 signaling. Neh2 and Neh6 domains of NRF2 participate in NRF2 degradation in response to redox and electrophile changes (KEAP1) and to signaling kinases, respectively. The Neh2 domain binds to the E3 ligase adapter KEAP1, which presents NRF2 for ubiquitination to the CUL3/RBX1 complex. The Neh6 domain requires previous phosphorylation by GSK-3β to bind to the E3 ligase adapter β-TrCP and subsequent ubiquitination by the CUL1/RBX1 complex. NRF2 signaling is activated in the presence of xenobiotics, reactive oxygen species (ROS)/electrophiles, or compounds like sulforaphane (SFN) or dimethyl fumarate (DMF) by the release from KEAP1 or β-TrCP binding and is translocated to the nucleus and binds to ARE consensus sequences, which allows for the activation of NRF2-dependent genes.

**Figure 2 antioxidants-12-01564-f002:**
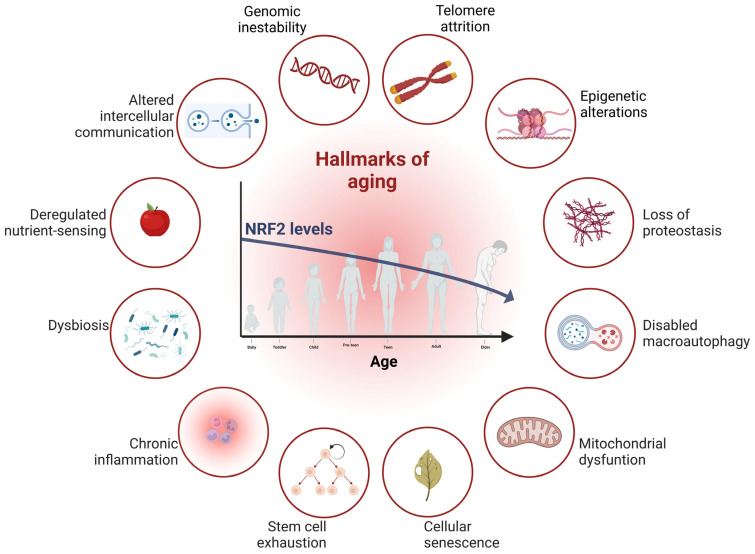
**Interconnections between aging and NRF2.** Scheme modified from [20] gathers the proposed 12 hallmarks of aging, in which involvement of the transcription factor NRF2 has been described. In this review, we will highlight the relationship between NRF2 and genomic instability, proteasomes, autophagy, mitochondrial dysfunction, cellular senescence, stem cell exhaustion, and inflammation.

**Figure 3 antioxidants-12-01564-f003:**
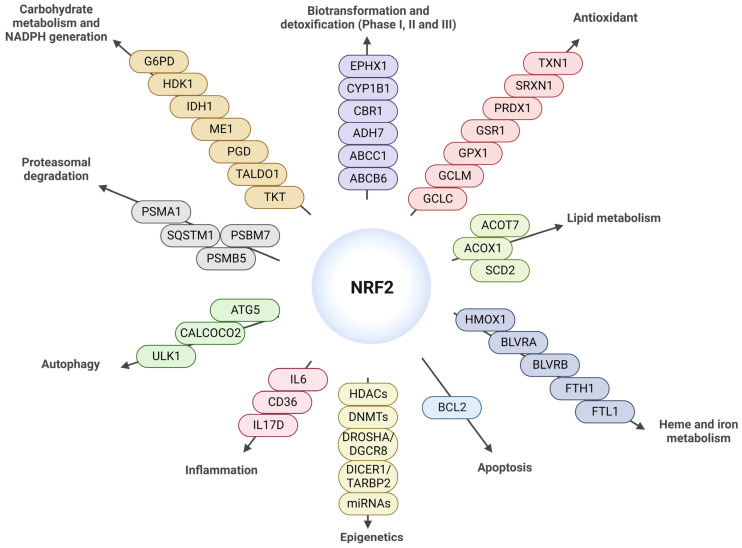
**Diagram of the main targets of NRF2** modified from [34]. NRF2 is implicated in the regulation of biotransformation and detoxification proteins (Phases I, II, III), antioxidants, lipid metabolism, heme and iron metabolism, apoptosis, epigenetics, inflammation, autophagy, proteasomal degradation, carbohydrate metabolism, and NADPH generation.

**Figure 4 antioxidants-12-01564-f004:**
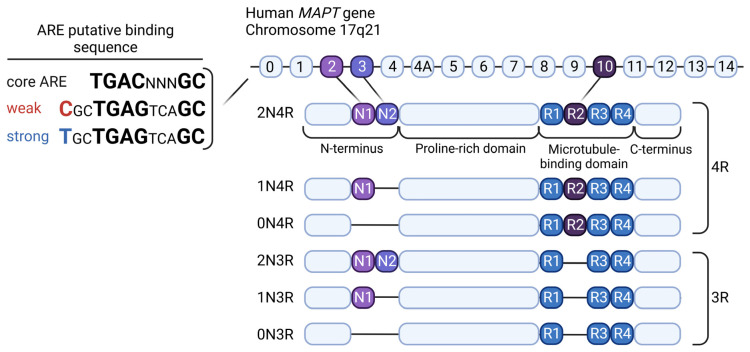
**Diagram of the TAU isoforms and ARE binding sequence** modified from [100,101], respectively. Six TAU isoforms are present in the human brain through different combinations of the splicing of exons 2, 3, and 10. The characteristics of the N-terminal domain, N1 and N2, are generated from exons 2 and 3, respectively. Exon 10 encodes the second microtubule-binding repeat domain, R2. Depending on the presence of the R2 domain, the TAU protein becomes either 3R or 4R TAU. In contrast, an ARE-binding SNP in the first intron of the *MAPT* gene that can confer weak (C) or strong (T) binding of NRF2 has been proposed.

**Figure 5 antioxidants-12-01564-f005:**
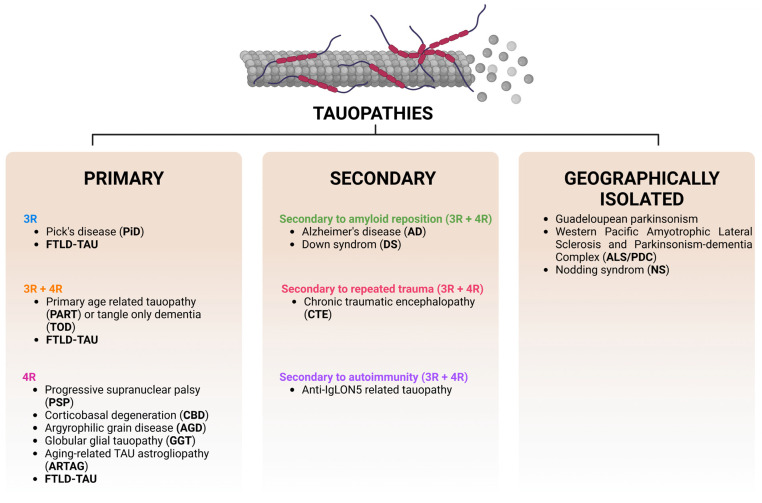
**Classification of tauopathies.** Tauopathies are divided into primary, secondary, or geographically isolated. Depending on the TAU isoform composing the intracellular inclusions, tauopathies can also be classified as 3R, 4R, and 3R + 4R tauopathies. Frontotemporal lobar degeneration–TAU (FTLD-TAU).

## Data Availability

Not applicable.

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
