# Peer review of "Aging, NRF2, and TAU: A Perfect Match for Neurodegeneration?"

_antioxidants, 2023, doi:10.3390/antiox12081564_

Round 1
Reviewer 1 Report
The review article entitled ‘Aging, NRF2 and TAU: a perfect match for neurodegeneration?’ by Brackhan et al. describes the interconnection between NRF2 and TAU, and the relevance of alterations in the NRF2 signaling pathway in tauopathies associated neurodegenerative diseases. There are a few areas that need more details and clarity, that will improve the quality of the manuscript:
1. The core of the review is based on the interconnection between NRF2 and TAU. The authors failed to describe how NRF2 connected with Tau. Even though the published articles not giving the exact mechanism, based on the available literature authors can make a proposed mechanism.
2. In Alzheimer’s disease, neuroinflammation (gliosis) starts before the Tau deposition, so interaction between Aβ and NRF2 is more important than Tau? Or is there any evidence that describes the connection between Aβ, Tau and NRF2?
3. Does the NRF2 and Tau interaction happen in the same cell (neuron) or in different cell types (neuron and glial cell)? If it happens in different cell types, how does the interaction occur? Explain in detail whether the mechanism occurs via paracrine/ juxtracrine signaling.
4. How does miRNA regulate NRF2 and thereby neuroinflammation?
5. It will be good to describe in detail, how NRF2 controls detoxifying enzymes in the brain, with the help of a figure, rather than the Figure 5.
6. How does the interconnection between Tau and NRF2 controls metabolism in the aging brain?
Author Response
Dear editor,
Thank you very much for your answer and your consideration of our review (antioxidants- 2530542). We think that the reviewers’ comments and suggestions can improve the message and the quality of our work. All editor’s and reviewer’s comments have been addressed as follows:
Reviewer 1
The review article entitled ‘Aging, NRF2 and TAU: a perfect match for neurodegeneration?’ by Brackhan et al. describes the interconnection between NRF2 and TAU, and the relevance of alterations in the NRF2 signaling pathway in tauopathies associated neurodegenerative diseases. There are a few areas that need more details and clarity, that will improve the quality of the manuscript:
- The core of the review is based on the interconnection between NRF2 and TAU. The authors failed to describe how NRF2 connected with Tau. Even though the published articles not giving the exact mechanism, based on the available literature authors can make a proposed mechanism.
ANSWER: Precisely the interconnection between NRF2 and TAU is a very interesting field yet to be explored, beyond the fact that TAU mRNA levels can be modulated by NRF2. For this reason, we have preferred to be cautious in exposing mechanisms that propose how both proteins are related. Experimental data obtained by our own research group demonstrated that neuronal overexpression of TAUP301L does not induce the NRF2 signaling pathway in these cells. But neuronal TAUP301L overexpression induces the expression and release of the chemokine fractalkine (CX3CL1), which at the CNS level acts on microglia, which express the CX3CR1 receptor. Activation of this receptor by CX3CL1 induces the activation of the NRF2 signaling pathway, with anti-inflammatory effects and on phagocytosis, an essential function of microglia. We do not know whether this effect is due to the P301L mutation or the overexpression of the TAU protein itself. But at the level of astrocytes or other cell types that are also altered by neuronal TAU expression, no studies have been performed. Following the reviewer's indications, we have highlighted this point in the section on the interconnection between NRF2 and TAU.
- In Alzheimer’s disease, neuroinflammation (gliosis) starts before the Tau deposition, so interaction between Aβ and NRF2 is more important than Tau? Or is there any evidence that describes the connection between Aβ, Tau and NRF2?
ANSWER: We agree with the reviewer that neuroinflammatory processes usually occur as an early event in the course of neurodegeneration. In relation to Alzheimer's disease, Aβ plays an essential role, and its possible relationship with NRF2 has been reviewed in numerous manuscripts. Most murine models of AD are based on mutations in APP and presenilin, and in these models the alterations that occur on the TAU protein are minimal. Therefore, there are triple murine models, which are the ones we have discussed in this review, and in which the involvement of NRF2 in the neurodegenerative process has been determined. We have not found more literature than we have described in this review. We have focused in this review on the TAU protein and the role of NRF2 in TAU-induced molecular alterations. This field, which is very broad because there are so many pathologies linked to TAU, has clearly not been explored in relation to NRF2.
- Does the NRF2 and Tau interaction happen in the same cell (neuron) or in different cell types (neuron and glial cell)? If it happens in different cell types, how does the interaction occur? Explain in detail whether the mechanism occurs via paracrine/ juxtracrine signaling.
ANSWER: We fully agree with the reviewer that this point is essential to understand how TAU and NRF2 converge in the molecular events that occur during the neurodegenerative process. And it is necessary to determine whether this "interaction" happens in the same cell types, to determine whether NRF2 modulation could be a beneficial strategy for tauopathies. Although we already made a brief mention of this in the review, in the new manuscript we will further discuss the importance of this fact.
- How does miRNA regulate NRF2 and thereby neuroinflammation?
ANSWER: Epigenetic modifications play an important role in the NRF2 pathway. Also, our own group has recently published that NRF2 has epigenetic regulatory functions that modulate HDACs, DNMTs and miRNA biogenesis. Although this is not a field we wanted to discuss in this review, we have included a small paragraph in the section on "NRF2 as an inflammatory modulator".
- It will be good to describe in detail, how NRF2 controls detoxifying enzymes in the brain, with the help of a figure, rather than the Figure 5.
ANSWER: The structure of this review is in relation to the recently described markers of aging and the involvement of NRF2 with them. We have been looking in the literature, and we have not seen that within these 12 markers are detoxifying enzymes. Therefore, although detoxification mechanisms are relevant in relation to NRF2, no specific paragraph has been written relating aging, NRF2 and detoxification enzymes. Nor have we seen how to fit a paragraph on how NRF2 controls detoxifying enzymes that would not modify the main thread of this review. Regarding figure 5, we believe it is important to keep it, especially for people who come from the NRF2 field, but do not know the classification of tauopathies.
- How does the interconnection between Tau and NRF2 controls metabolism in the aging brain?
ANSWER: Although there is no literature on the subject, we have attempted to address this point in the new manuscript.

Reviewer 2 Report
In the present manuscript (ID: antioxidants-2530542), Brackhan et al., highlight the physiopathological link between the aging, the stress-induced NRF2 transcription factor and the tau protein. Although the translational relevance of this review in the treatment of neurodegenerative is interesting, in my humble opinion the ms is not comprehensive in its present version and some important issues need still to be addressed/completed prior to be considered for publication on high-impact factor journal such as Antioxidants.
Major revisions:
1.In the second paragraph, the authors do not describe the other mechanisms of Keap1- independent regulation of Nrf2. In order to make the reading clear and exhaustive , this aspect should be included in the manuscript.
2.Another important aspect that authors do not address is the crucial, direct and/or indirect, role of NRF2 on metabolism, in particular on mitochondria, whose dysfunction is considered to play a large role in aging and neurodegenerative diseases such as Alzheimer's disease (AD) and other tauopathies. Please, include considerations in the manuscript.
3.NRF2 plays a key role in Ferroptosis, a form of neuronal death involved in AD, tau protein is associated with brain iron metabolism.The authors should also discuss this important topic in the manuscript.
4.A paragraph should be devoted to the therapeutic agents with modulation ability of NRF2 currently exploited for the cure of age-associated neurodegenerative diseases, including AD .
Moderate editing of English.
Author Response
Dear editor,
Thank you very much for your answer and your consideration of our review (antioxidants- 2530542). We think that the reviewers’ comments and suggestions can improve the message and the quality of our work. All editor’s and reviewer’s comments have been addressed as follows:
Reviewer 2
In the present manuscript (ID: antioxidants-2530542), Brackhan et al., highlight the physiopathological link between the aging, the stress-induced NRF2 transcription factor and the tau protein. Although the translational relevance of this review in the treatment of neurodegenerative is interesting, in my humble opinion the ms is not comprehensive in its present version and some important issues need still to be addressed/completed prior to be considered for publication on high-impact factor journal such as Antioxidants.
Major revisions:
1.In the second paragraph, the authors do not describe the other mechanisms of Keap1- independent regulation of Nrf2. In order to make the reading clear and exhaustive, this aspect should be included in the manuscript.
ANSWER: In the original manuscript we have described the two main mechanisms of NRF2 regulation: KEAP1 and GSK-3, which are detailed in Figure 1. We are aware that there are other mechanisms of modulation of the NRF2 pathway, mainly by different kinases. We have included a paragraph to note the involvement of other kinases in NRF2 modulation.
2.Another important aspect that authors do not address is the crucial, direct and/or indirect, role of NRF2 on metabolism, in particular on mitochondria, whose dysfunction is considered to play a large role in aging and neurodegenerative diseases such as Alzheimer's disease (AD) and other tauopathies. Please, include considerations in the manuscript.
ANSWER: We fully agree with the reviewer that mitochondria play a key role not only in TAU-induced neurodegeneration, but in almost all neurodegenerative diseases. For this reason, in principle we had not included it in the review. But thanks to the reviewer, we have realized that it cannot be left unmentioned, and have therefore included a new section in the manuscript.
3.NRF2 plays a key role in Ferroptosis, a form of neuronal death involved in AD, tau protein is associated with brain iron metabolism. The authors should also discuss this important topic in the manuscript.
ANSWER: The field of ferroptosis is emerging on the horizon of neurodegenerative diseases. Although there is still not much literature on TAU protein, i.e. we have found literature mainly related to AD and only one paper related to a primary tauopathy that has no DOI, we will include a paragraph about it in the new version of the manuscript.
4.A paragraph should be devoted to the therapeutic agents with modulation ability of NRF2 currently exploited for the cure of age-associated neurodegenerative diseases, including AD.
ANSWER: Due to the potential therapeutic use of NRF2 in the different tauopathies, we had already included a paragraph in each case. Unfortunately, apart from TAU-dependent frontotemporal dementia, there are no studies on the potential use of NRF2 inducers as a therapeutic strategy in other primary tauopathies. All that exists in the literature is that it is thought to have a potential use, without studies to support it. In relation to AD, this disease is a secondary tauopathy, i.e. there are many other factors involved in the development of molecular alterations, such as Aβ. Although AD is the main cause of dementia and there is much evidence that NRF2 modulation would be beneficial, this point is out of focus of this review. There are numerous studies relating the therapeutic efficacy of NRF2 modulators in AD, so this point would be a review in itself. In the AD section we have included a short paragraph supporting the potential use of NRF2 modulators in AD.

Reviewer 3 Report
This is a well written, informative and balanced review that just requires a few minor edits, including language.
1. Title "a perfect match" for neurodegeneration sounds like something positive, but neurodegeneration is a disease. I would rather suggest something like "Partners in crime" or similar. Please consider.
2. line 77: The definition of aging is not at the cellular level (that would be senescence) but at the physiological level.
3. Please give the abbreviation for "sMAF" (line 262)and also describe CBD and PSP not only in figure 5 but also in the text.
4. line 125: Please explain what you mean with "biotransformation"
5. line 136: please specify "aggregation" by listing the relevant proteins.
6. lines 162, 171 and 176 require a reference each.
7. line 178: Reference 60 also mentions self renewal in addition to controlling pluripotency as proteasomal regulator. This is different from a "pluripotency genes" which seems over-interpreted and not entirely correct.
8. In line 186 you probably refer to the ratios of the different neuronal cell types. Please rephrase as cells ae not "produced".
9. Importantly, in line 239: There is not such a thing as "accumulation" of oxidative stress since ROS are VERY short-lived and thus cannot accumulate, only the resulting damage can. Please correct!
10. Lines 276-282, line 299 and throughout the manuscript: Please use the term "expression" only for genes, not for RNA or proteins. rather use "level/amounts" here.
11. line 314: please give a reference. Also, more than 30% seems rather substantial as it is one third and not negligible. Perhaps rephrase.
12. line 380: please explain "bv" in front of FTD.
13. line 404: What do you understand under "reminiscent cells"? Consider rephrasing.
14. Heading 6.3. You mention alpha-synuclein only very briefly in the beginning of this heading. Are there no studies on NRF2 and asyn in these 2 diseases? Perhaps include them as well.
15. line 473: as you seem to refer to enzymes=proteins, do not use the term "expression" )see also comment above). Likewise in lines 479 and 484: RNA is not expressed, just genes are!
16. line 471: The term "mild AD" is not very scientific-better use "low Braak stages".
17. line 487: Proteins are LOCALISED in the different subcellular compartments, NOT expressed!
18. line 496: Why does a low mitochondrial respiration is associated to low ROS generation? Since mito respiration CAUSES ROS it should be correlated to the levels of generation. If you refer to non-mitochondrial sources (NOXes) please state or if you mean just ROS LEVELS due to antioxidant scavenging, that please do not refer to "production/generation" as this is incorrect.
language: line 58: remove "the". line 93: The end of the sentence "also concerning TAU protein" does not fit the rest of the sentence which lists various processes. Please rephrase. line 147: replace "with" with "in". line 163: add "the" in front of "Keap" and it should be "THESE fibroblastS" since there are more than 1 cell in a culture. line 173 "in what is SASP" is wrong grammar/language. Please rephrase. line 198: I think It should be "phagocytozing". line 217 add "the" in front of "small". line 218: "THIS evidence concverges" "evidence" is used in singular. line 244 : please replace "precisely" with "In particular". line 252 and others : please replace "produced" by "generated". "Production" is a more technical term. line 266: You cannot use "THey" since you never mentioned any authors before. Please rephrase, for example stating "The study" instead. line 303: "Divided IN" not "as". line 343: replace "produce" with "results in". line 396: "On one hand", no article. line 400: remove comma. line 417: remove "at" before "postmortem".
The English is in general very good, just some very minor corrections are required which I mentioned to the authors.
Author Response
Dear editor,
Thank you very much for your answer and your consideration of our review (antioxidants- 2530542). We think that the reviewers’ comments and suggestions can improve the message and the quality of our work. All editor’s and reviewer’s comments have been addressed as follows:
Reviewer 3
This is a well written, informative and balanced review that just requires a few minor edits, including language.
- Title "a perfect match" for neurodegeneration sounds like something positive, but neurodegeneration is a disease. I would rather suggest something like "Partners in crime" or similar. Please consider.
ANSWER: We fully understand the point made by the reviewer. However, we have several presentation titles in which we have written "partners in crime" that we wanted to make a change. Hence, we wrote "perfect match", but not as something positive, but as a perfect combination that produces the appearance of diseases. Another possible title would be: “Aging, NRF2 and TAU: conspiracy to neurodegeneration?”.
Unfortunately, the lines indicated by the reviewer do not match the lines we have in the manuscript. So, we have made every effort to find what the reviewer indicates, although we have not been able to find them in all cases.
- line 77: The definition of aging is not at the cellular level (that would be senescence) but at the physiological level.
ANSWER: In this review we have not provided a definition of aging. We have indicated that during aging, several events occur at the molecular level, as stated in the new review by López-Otín.
- Please give the abbreviation for "sMAF" (line 262)and also describe CBD and PSP not only in figure 5 but also in the text.
ANSWER: We have included it in the new version of the manuscript.
- line 125: Please explain what you mean with "biotransformation"
ANSWER: We have included the definition of biotransformation in the new version of the manuscript.
- line 136: please specify "aggregation" by listing the relevant proteins.
ANSWER: Depending on the context/pathology, different proteins may be involved in the aggregation processes but in relation to this review, we mainly discuss TAU aggregation. Due to its structure, it is a protein that is prone to aggregate, and at the beginning it does not need other proteins to do so. The question is whether, as time progresses, these aggregates become a sink where other proteins also converge. In this review we have not addressed this point, but simply that TAU, under certain circumstances, is given to aggregate by itself.
- lines 162, 171 and 176 require a reference each.
ANSWER: We have included it in the new version of the manuscript.
- line 178: Reference 60 also mentions self renewal in addition to controlling pluripotency as proteasomal regulator. This is different from a "pluripotency genes" which seems over-interpreted and not entirely correct.
ANSWER: We have included a sentence to try to explain it better. In fact, it is the authors themselves who indicate that NRF2 acts as a key pluripotency gene in human embryonic stem cells and we do think that they have demonstrated it in the published work.
- In line 186 you probably refer to the ratios of the different neuronal cell types. Please rephrase as cells ae not "produced".
ANSWER: Due to the fact that we do not correlate the line numbers indicated, and in what the reviewer has written we cannot identify it in the text, we have not been able to include a change/improvement in the text.
- Importantly, in line 239: There is not such a thing as "accumulation" of oxidative stress since ROS are VERY short-lived and thus cannot accumulate, only the resulting damage can. Please correct!
ANSWER: we fully agree with the reviewer, and we have modified it so that it is expressed correctly: the accumulation of the damage produced by oxidative stress occurs.
For example, this appears in our text on line 207, but we could have searched for "accumulation of oxidative stress".
- Lines 276-282, line 299 and throughout the manuscript: Please use the term "expression" only for genes, not for RNA or proteins. rather use "level/amounts" here.
ANSWER: we have corrected this point in the text
- line 314: please give a reference. Also, more than 30% seems rather substantial as it is one third and not negligible. Perhaps rephrase.
ANSWER: We believe the reviewer is referring to the sentence "In general, most cases of tauopathies are sporadic and only approx. 31% have been reported as a family form (with variations within the different tauopathies)." In our case, line 270. We agree with the reviewer. We believe that the word "only" is superfluous, so we have removed it and included a supporting reference.
- line 380: please explain "bv" in front of F TD.
ANSWER: The FTLD section, when describing the different types, includes the definition of bv. Our line 358.
- line 404: What do you understand under "reminiscent cells"? Consider rephrasing.
ANSWER: If it is the sentence: These data suggest that although there is a significant global decrease of NRF2 in the frontal cortex, reminiscent cells continue to activate NRF2 in response to oxidative stress [130], we change reminiscent by remaining.
- Heading 6.3. You mention alpha-synuclein only very briefly in the beginning of this heading. Are there no studies on NRF2 and asyn in these 2 diseases? Perhaps include them as well.
ANSWER: In both PSP and CBD there are no α-synuclein alterations, but rather tauopathies. I have been searching in the literature, and indeed no αsynuclein alterations have been described in these pathologies, although their clinical pictures are parkinsonian. Both are primary tauopathies of type 4R. Thus, there are no studies of α-synuclein and NRF2 in PSP and CBD.
- line 473: as you seem to refer to enzymes=proteins, do not use the term "expression" )see also comment above). Likewise in lines 479 and 484: RNA is not expressed, just genes are!
ANSWER: Thank you for your comment and we will proceed to make the appropriate changes.
- line 471: The term "mild AD" is not very scientific-better use "low Braak stages".
ANSWER: We have introduced the appropriate changes in the new version of the manuscript.
- line 487: Proteins are LOCALISED in the different subcellular compartments, NOT expressed!
ANSWER: We have included the change in the new manuscript.
- line 496: Why does a low mitochondrial respiration is associated to low ROS generation? Since mito respiration CAUSES ROS it should be correlated to the levels of generation. If you refer to non-mitochondrial sources (NOXes) please state or if you mean just ROS LEVELS due to antioxidant scavenging, that please do not refer to "production/generation" as this is incorrect.
ANSWER: Because more studies are necessary, we have changed the different phrases that allude to ROS.
language: line 58: remove "the".Done. line 93: The end of the sentence "also concerning TAU protein" does not fit the rest of the sentence which lists various processes. We have removed it. Please rephrase. line 147: replace "with" with "in". We haven't found it. If you can be more specific? Line 163: add "the" in front of "Keap" and it should be "THESE fibroblastS" since there are more than 1 cell in a culture. Done. line 173 "in what is SASP" is wrong grammar/language. Please rephrase. We have changed it. line 198: I think It should be "phagocytozing". We have found in several publications "phagocytizing" and that is why we have written it like this. line 217 add "the" in front of "small". Done. line 218: "THIS evidence concverges" "evidence" is used in singular. Done. line 244 : please replace "precisely" with "In particular". Done. line 252 and others : please replace "produced" by "generated". "Production" is a more technical term. Done. line 266: You cannot use "THey" since you never mentioned any authors before. Please rephrase, for example stating "The study" instead. Done. line 303: "Divided IN" not "as". Done. line 343: replace "produce" with "results in". line 396: "On one hand", no article. Done. line 400: remove comma. line 417: remove "at" before "postmortem". Done.

Round 2
Reviewer 2 Report
The authors addressed all the suggestions of reviewer.